# A Diarrhoeagenic *Enteropathogenic*
*Escherichia coli* (*EPEC*) Infection Outbreak That Occurred among Elementary School Children in Gyeongsangbuk-Do Province of South Korea Was Associated with Consumption of Water-Contaminated Food Items

**DOI:** 10.3390/ijerph17093149

**Published:** 2020-04-30

**Authors:** Min-A Lim, Ji-Yeong Kim, Dilaram Acharya, Bishnu Bahadur Bajgain, Ji-Hyuk Park, Seok-Ju Yoo, Kwan Lee

**Affiliations:** 1Division of Public Health Policy, Gyeongsangbuk-Do Provincial Government, Andong 36759, Korea; minadid@korea.kr; 2Sangju-si Public Health Center, Sangju 37183, Korea; end1309@korea.kr; 3Department of Preventive Medicine, College of Medicine, Dongguk University, Gyeongju 38066, Korea; dilaramacharya123@gmail.com (D.A.); skeyd@naver.com (J.-H.P.); medhippo@dongguk.ac.kr (S.-J.Y.); 4Department of Community Medicine, Kathmandu University, Devdaha Medical College and Research Institute, Rupandehi 32907, Nepal; 5Department of Community Health Sciences, University of Calgary, Calgary, AB T2N 1N4, Canada; bishnu.bajgain@ucalgary.ca

**Keywords:** diarrhoea, *enteropathogenic Escherichia coli* outbreak, epidemiology, school feeding

## Abstract

(1) Background: In response to the notification made by an elementary school authority that reported a number of elementary school children being absent in three schools as a result of gastroenteritis symptoms on 4 July 2018, in Gyeongsangbuk-Do Province, South Korea, an epidemic investigation was carried out to determine the extent, cause, and source of the outbreak in order to prevent secondary cases and make recommendations to prevent future recurrences. (2) Methods: In this epidemiologic study, a total of 106 human subjects (school children, staff members, and cooks) who had consumed the possibly contaminated foodstuffs were enrolled retrospectively. Human specimens from clinically defined cases, food and drinks, supply and storage of them, and environmental and sanitary conditions were also assessed by observation, laboratory tests, and survey questionnaires—where and whatever applicable. The attack rate and positive rate for human specimens were first presented followed by the calculation of the relative risk ratio (RR) with 95% CI (confidence intervals) in order to identify the exposure and outcome relationships. (3) Results: The attack rate was 12.26% (13/106) for those who had ingested the food items at the three schools and the positive rate of enteropathogenic *Escherichia coli* (*EPEC*) was 15.38% (2/13). The relative risk (RR) of developing food poisoning of those who consumed the cucumber chili with ssamjang and seasoned cucumber and chives were 4.55 (95% CI 1.05–19.54) and 9.20 (95% CI 1.24–68.22), respectively. In addition, within the human specimens as well as the water and environmental samples different strains of diarrhoeagenic *enteropathogenic Escherichia coli* (*EPEC*) were detected. (4) Conclusions: Provision of safe and wholesome water access to all elementary schools by concerned authorities, especially during the likely seasons of water source contamination, as well as health education promotion about foodborne outbreaks to all school stakeholders is therefore recommended.

## 1. Introduction

Food and waterborne illnesses are worldwide-prevalent public health problems [1,2]. Food poisoning is an acute gastroenteritis caused from ingestion of food or drinks with either microbial or non-microbial contamination. Acute gastroenteritis is one of the leading causes of morbidity and mortality [2]. The World Health Organization (WHO) estimated that every year 600 million (almost 1 in 10 people) fall in sick and nearly 420,000 deaths occurs worldwide as a result of contaminated food consumption, giving rise the loss of 33 million healthy lives (DALYs) [3]. Out of this number, children under 5 years of age hold 40% of the morbidity and 125,000 mortalities every year. A report shows that more than 200 diseases ranging from diarrhoea to cancer are caused due to food poisoning that can lead to losses in national economy, increased health care cost, and impede the development of trade and tourism [3]. Likewise, contaminated water consumption is one of the major causes of waterborne human infections [4]. Waterborne diarrhoeal disease claims 2 million deaths worldwide each year, mostly in children below 5 years of age. Of the total world population, only approximately 663 million people are reported to have consistent access to improved drinking water sources [4,5].

The major features of food poisoning are headache, intense thirst, acute vomiting, giddiness, diarrhoea, slow pulse, cramps and rigors, colicky pain, cold, and clammy skin [2]. *Salmonella, Campylobacter, Vibrio cholerae, and enterohaemorrhagic Escherichia coli* are the most common foodborne pathogens. The usual sources of these pathogen are raw or undercooked foodstuffs, such as egg, meat, milk or milk products, fruits and vegetables, and water [3]. Equally, several waterborne gastroenteritis epidemics have been found to be generated by diarrhoeagenic *Escherichia coli* (DEC), *Gram-negative bacteria*, which are divided into five groups: *enterohaemorrhagic E. coli (EHEC), enteropathogenic E. coli (EPEC), enterotoxigenic E. coli (ETEC), enteroaggregative E. coli (EAEC),* and *enteroinvasive E. coli (EIEC)* [6,7]. 

Food poisoning and waterborne outbreaks in non-community places, such as schools, colleges, restaurants, and hotels, are found to be linked with several microbial and non-microbial factors, likely due to the contamination of food and drinks. Low pressure conditions of water, low residual chlorine concentration, and broken-down piped water supply are responsible for the intrusion of pathogenic microorganisms in water, while food-related factors, such as food handling policy, undercooked, improper labelling, and cleaning of raw foodstuffs, can provide an opportunity for microbes to invade into the food items [3,8,9].

Among many other sources, the chief sources of water, food, and drinks contamination usually reported are human sewage, animal wastes, overly used toxic agricultural chemicals, and pesticides that are mixed up into the water supply system [3,4,10]. Removal of biological and chemical contamination from drinking water prior to consumption is a crucial step in water safety [11]. Managing the quality of water from all sources, including public water and groundwater, by a multistep treatment system can reduce the risk of infection in the human body [12]. Similarly, standard food safety policy, appropriate handling, proper cooking, cleaning, training to food handlers, as well as good coordination between governments, producers, and consumers help ensure food safety [3].

Although there are infrequent reports from developed countries about the outbreak of food and waterborne infections associated with *EHEC* infections as the etiological agent, multiple bacterial and viral strains may have been involved for such outbreaks [3,7]. Despite having less attention to *EHEC* infections occurring as a result of consumption of contaminated food and water in developed countries, especially in non-community settings, such findings are important for the evidence-based strategy formulation in order to reduce and prevent recurrence of diarrhoeal illnesses. This study aimed to identify the extent, cause, and source of an outbreak—through epidemic investigation—that occurred among elementary school children in Gyeongsangbuk-Do Province, South Korea, to prevent secondary cases and to make recommendations as to the prevention of future recurrences.

## 2. Materials and Methods

### 2.1. Study Background and Settings

The epidemiological investigation was carried out with the receipt of the notification from a school authority that a number of elementary school children were absent in school as a result of symptoms of gastroenteritis diarrhoea, fever, and abdominal pain on 4 July 2018, in Gyeongsangbuk-Do Province, South Korea. Administratively, South Korea is comprised of 17 first-tier administrative divisions: 6 metropolitan cities, 1 special city, 1 special autonomous city, and 9 provinces, including one special autonomous province. These are further subdivided into a variety of smaller entities, including cities (Si), counties (Gun), districts (Gu), towns (Eup), townships (Myeon), neighbourhoods (Dong), and villages (Ri) [13]. Of the 106 human subjects who consumed the possibly contaminated foodstuffs in three elementary schools (one main elementary and another two branches) from the city (Si) of Gyeongsangbuk-Do Province of South Korea, 13 developed food poisoning (elementary school children and teachers). The study team consisted of epidemiologists, laboratory personnel to collect sample specimens from the affected subjects, and a food hygienist to check and collect the suspected food and drinks causing the food poisoning, as well as environmental samples.

### 2.2. Epidemiological Investigation

A case of food poisoning was defined as those who were having 3 episodes of diarrhoea within 24 h or 2 diarrhoeal episodes, and one or more symptoms of nausea and vomiting, abdominal pain and fever, as per the guideline suggested by Korea Centre for Disease Control and Prevention (KCDC) [14]. Similarly, the cases of food poisoning were confirmed with cultures positive for *EPEC*. A total of 13 cases occurred during the period of 26 June to 3 July 2018 and were analysed retrospectively to identify the risk factors associated with food and drinks and environmental stuffs. All relevant study subjects (school children and their parents, school teachers, cook workers, and food distributors) were interviewed using in-built epidemiological case-sheets as suggested by KCDC to collect a wide range of information about the biosocial aspects as well as the food and drinks consumed, processed, prepared, and delivered, and exposure to other environmental and ecological conditions. All the cases developed during the study period were monitored and supervised continuously and follow-up surveillances were conducted twice during the incubation period until the study settings were declared free of disease. In the containment of the outbreak and its prevention to further spread, a number of general and specific measures were applied, such as imparting health education about water and foodborne infections and methods of prevention and control like the use of boiling water and proper hand washing to all of the target population, including users of tap water in neighbouring villages. In addition, to prevent further occurrences, disinfection was carried out in school lunch facilities and restrooms, and the water and sewage establishments were suggested to liaise with the school and the village’s waterworks management.

### 2.3. Environmental and Ecological Assessment

A study team examined all the possible environmental and ecological factors to identify the causes of outbreak. In this course of action, the food hygiene officer examined the ingested and stored food items at the school meal centre, and checked for drinking and cooking water, tap water and groundwater in the food service centre, and the chlorine concentration. Moreover, the interior of the kitchen and the inspection status of the cooking chamber were also observed. Food, water, environmental, and other ecological samples were collected and tested to detect the microbes causing the outbreak.

### 2.4. Microbiological and Biochemical Examination of Human and Environmental Samples

The Gyeongsangbuk-do Health Environment Laboratory technologists collected clinical sample specimen from all cases affected. Rectal swabs and stool samples were collected. Thus, collected samples were cultured and tested at the local health authority. Further, these isolates were transferred to the KCDC in order to confirm the causative agents. The food poisoning-causing pathogens tested for were bacteria (16 species): *Cholera, Salmonella typhi, Salmonella paratyphi, Shigellosis, enterohemorrhagic E. coli, Salmonella spp., Vibrio parahaemolyticus, enterotoxic E. coli, enteroinvasive E. coli, enteropathogenic E. coli, Campylobacter jejuni, Clostridium perfringens, Staphylococcus aureus, Bacillus cereus, Yersinia enterocolitica, Listeria monocytogenes*; and *viruses* (6 species): Hepatitis A, Rotavirus, Astrovirus, Adenovirus, Norovirus (G1, G2), and Sapovirus. In case of food and drinks and environmental specimens, the following possible pathogens and biochemical investigation were performed: Drinking water (9 kinds): general bacteria, total coliform group, faecal coliform group, ammonia–nitrogen (NH_3_-N), nitrate–nitrogen (NO_3_-N), free residual chlorine, potassium permanganate consumption, chlorine ion, and sulfuric acid ion.

### 2.5. Statistical Analyses

The Chi-square test to investigate the different food and drinks and environmental risk factors was employed. Thus, between exposure and non-exposure cohorts, the relative risk (RR) with 95% confidence interval (CI) for all samples examined were calculated. A *p*-value < 0.05 was set as statistically significant. All statistical analyses were conducted using SPSS for Windows (version 22.0, SPSS Inc. Chicago, IL, USA).

### 2.6. Ethics

This study falls within the rules and regulations of the Korean Bioethics and Safety Act (Article, 15) and Infectious Disease Control and Prevention Act (Article 18) set by the Korea Center for Disease Control and Prevention, Ministry of Health and Welfare, and the Provincial Government of South Korea; therefore, an additional ethical approval from the institutional review board or ethical committee was not necessary. All personal details were removed prior to data analysis.

## 3. Results

### 3.1. Descriptive Results

The incidence rate according to case definition was 12.26% (13/106) of those who had ingested the food items at three elementary schools and the positive rate of *EPEC* was 15.38% (2/13). Among the clinically defined cases, 9/13 (69.23%) were male, while the laboratory-confirmed two cases were one male and another female. Of total 13 defined cases, the percent reported to have diarrhoea followed by fever was 76.9%, abdominal pain 53.8%, nausea in 15.4%, and the lowest percentage was chills (7.7%). Figure 1 shows that the trend of the cases among the study subjects in the study area. Of the 13 cases, nearly half (46.1%) of the cases occurred after the third day of exposure (within the one incubation period), resembling a point-source epidemic.

### 3.2. Results of Laboratory Tests of Human Specimens

Table 1 shows the laboratory test results of the human specimens. Of the 13 clinically confirmed cases of food poisoning, 15.38% (2/13) were confirmed to have *EPEC* O139, ONT in the human specimens; both of them had ingested the possibly contaminated foodstuffs.

### 3.3. Diarrhoeagenic EPEC Infection Outbreak Associated with Consumption of Water-Contaminated Food Items

Of number of food items consumed that were obtained through personal interviews of the affected subjects, the cucumber tofu that was supplied in the meals was found to be associated with the food poisoning (Table 2). The relative risk (RR) of developing food poisoning of those who consumed the cucumber chili with ssamjang on 27 June and seasoned cucumber and chives on June 28 were 4.55 (95% CI 1.05–19.54) and 9.20 (95% CI 1.24–68.22), respectively.

### 3.4. Results of Laboratory Tests of Environmental Specimens

Table 3 demonstrates the 16 types of food poisoning-causing bacteria and nine types of water-borne bacteria tested for in 39 food items, as well as other environmental and ecological specimens. It was revealed that the cooking water (from the village water supply) and dishwashing water (from the village water supply) were found to have *EPEC* O87. Regarding the concentration of the coliforms, both the cooking and drinking water (from the village water supply) had both total coliforms (+) and faecal coliforms (+), while the main school water purifier (from the village water supply) had only total coliforms (+). 

## 4. Discussion

This outbreak investigation sought epidemiologic, laboratory and environmental evidences in the occurrence of the outbreak of diarrhoeagenic *EPEC* infection in three elementary school’s children in the Gyeongsangbuk-Do province of South Korea. We observed the clinically defined cases [14]. (those having at least three episodes of diarrhoea within 24 h or two and more diarrhoeas and one or more symptoms of nausea, diarrhoea, and fever as defined by the KCDC) of those who had consumed foods served at schools from 25 June to 29 June 2018. The attack rate among clinically confirmed cases was 12.26% and the laboratory confirmed cases with *EPEC* was 15.38%, while nearly half (46.1%) of those of clinically confirmed cases occurred on the third day of exposure (within the one incubation period), indicating that the outbreak is a point-source epidemic. Previous similar water and foodborne outbreaks caused by *EPEC* occurred in similar school settings of South Korea [15,16], which reported more than twice higher attack rates compared with this study; all these studies, including the current one, suggest that *EPEC* outbreaks in South Korean school settings should not be underestimated. In fact, Intestinal pathogens, such as *enteropathogenic E. coli* (*EPEC*) and *enterotoxigenic E. coli* (ETEC) are less common in developed countries, and are usually endemic in developing countries [17], and therefore, less attention is given in developed countries to such pathogens unless the diarrhoeagenic outbreak is more obviously prevalent and reported. Accordingly, in South Korea, diarrheal diseases caused by *EPEC* has not been listed as a national notifiable disease. However, the reason why it is important in terms of public health significance in South Korea, is that it is difficult to recognize unless there is massive prevalence of diarrhoeal diseases, and that of the symptoms of *EPEC* infections are reported to be weaker and illnesses are sporadic compared to other diarrhoeal illnesses [18]. In addition, since gastrointestinal illnesses occurring as a result of faulty water supply systems is on the decline in developed countries, it has been of less concern and thus overlooked in conditions favouring water contamination, especially in rural settings during rainy seasons [19].

In our study, the human cases specimen investigation confirmed *EPEC* O139, ONT strains in those clinically confirmed cases who were exposed to foods served at school. Of the 29 foodstuffs consumed during the period of 25 June to 29 June 2018, cucumber chili with ssamjang and seasoned cucumber and chives posed a higher relative risk (RR) of giving food poisoning. The RR for developing food poisoning of those who consumed the cucumber chili with ssamjang on June 27 and Seasoned cucumber and chives on 28 June were 4.55 and 9.20, respectively, compared to their counterparts of non-exposure. Further, our environmental and ecological investigation revealed that cooking water and dishwashing water (from village water supply system) used in elementary schools were found to have *EPEC* O87. 

In South Korea, we generally believe that the centrally controlled water supply system is well established like in other developed countries, but in rural areas where there is no centrally controlled water supply network, and the stability of the simplified wide-area water supply needs to be managed. In recent years, even in the case of a simplified wide-area water supply, an automatic chlorine injector is installed and well maintained, but it needs be checked and managed periodically, and would need to be especially well managed when there is heavy rainfall over a long period, as well as during a draught that can lead to the reduction in efficacy of the chlorine used in the water supply system. In this investigation, during the period of 26~27 June, 2018, there was heavy rainfall all around the current settings, which we assume might have contaminated the source of the drinking water and lessened the chlorine concentration in the piped water supply system. In the current study settings, the foodstuffs were prepared in one main elementary school and it was supplied to the other two branches of the elementary schools. Thus, the same foodstuffs were served during the stated period to all three schools, and the water supply systems were different one another, but cases were observed in all three schools; it could, therefore, be logically assumed that the foodstuffs could have been prepared using contaminated water. Thus, considering the type of food exposure, the incubation period, recommended case definition, human specimen report, and environmental laboratory investigation, we logically concluded that the diarrhoeagenic outbreak that occurred among those three elementary school children was caused by *EPEC* infection as a result of water-contaminated foodstuffs consumption [20].

Although investigators attempted with their greater strengths to explore supportive evidence of the *EPEC* food poisoning, the study should have been considered in the light of some specific limitations. First, our retrospective analysis of the data obtained could not rule out the lower positivity test results of the human specimens despite all school personnel having consumed the same foodstuffs. This could be explained in that the affected individuals had already been treated with antibiotics before laboratory investigation started as the school authority reported the event lately because of weekend holidays. Another reason for the higher negative test results in the bacterial culture of the human specimens could be the possibility that the infective period had already been passed out, or the bacteria might not have been cultivated. This is supported by the fact that the intestinal bacteria decrease rapidly as the symptoms improve [18]. In line with the support of this logic, other studies reported positive test result of diarrhoeagenic bacteria among children with diarrhoea ranging from 2.7% to 16%, while it was at 4.0% in children without diarrhoea [21,22]. Secondly, although two of the foodstuffs (the cucumber chili with ssamjang and seasoned cucumber and chives) consumed posed a statistically significant higher RR of causing an outbreak, the same food sample could not be confirmed in culture due to the lack of appropriate food samples (in South Korea, food items can customarily be preserved up to 6 days and are in freeze-dried conditions) because the event was reported late, only on the 4th of July 2018, and outbreak investigation initiated on July 5. In this regard, it should be well advocated to all school stakeholders to notify any of the food- or waterborne illnesses at the soonest possible time to the concerned local health authority, which can help with robust investigation and initiate early interventions. Thirdly, this study might have suffered from recall bias as the data obtained were self-reported, and this may lead to misclassification of exposure. However, the authors attempted to capture the information through the use of already-piloted epidemiological case sheets recommended by the KCDC with different sets of questionnaires for school children, their parents, and the school stakeholders. Moreover, the study could be meaningful to underline how children are the most vulnerable group; a scientific study revealed a strategic point to better understand their health needs and to monitor the quality of care provided to this vulnerable population [23].

## 5. Conclusions

This epidemic investigation identified that the diarrhoeagenic *enteropathogenic Escherichia coli* infection outbreak that occurred among elementary school children was due to the consumption of water-contaminated cucumber chili with ssamjang and seasoned cucumber and chives food items. In addition, the human, water, and environmental samples tested detected the strains of diarrhoeagenic *enteropathogenic Escherichia coli*. South Korean central and provincial governments should ensure safe and wholesome water access to all elementary schools, especially during the likely seasons of water contamination, and the stakeholders of the elementary schools should have been well-advocated with regard to the health education promotion about such water- and foodborne outbreaks, as well as to the importance of early notification of such outbreaks so as to place effective interventions at the incipient stage and prevent future recurrences.

## Figures and Tables

**Figure 1 ijerph-17-03149-f001:**
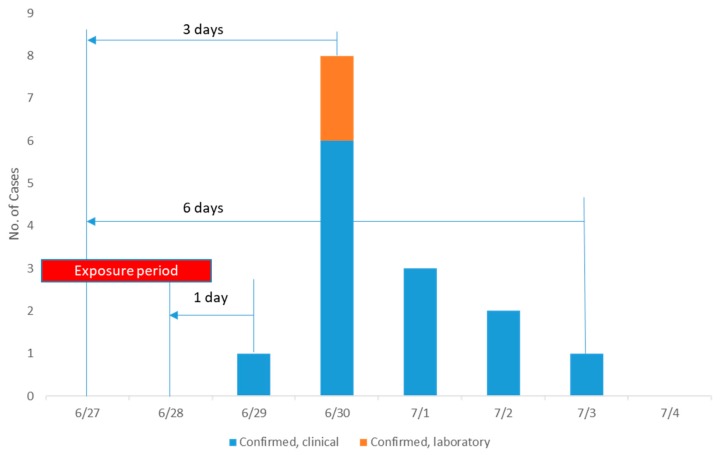
Distribution of food poisoning in clinically confirmed and laboratory-confirmed cases, considering the incubation period.

**Table 1 ijerph-17-03149-t001:** Laboratory test results of the human specimens.

Division	No. of Samples	Test Result (%)	Bacterial Strain
Cases	13	2 (15.38)	*EPEC* O139, ONT
Cook worker	4	Not detected	
School students participated in distributing food *	3	2 (66.7)	*EPEC* O170, ONT
A food teacher participated in distributing food	2	Not detected	
Food delivery driver	1	Not detected	

* Asymptomatic cases: usually laboratory tests are conducted on the cases with symptoms, but the tests were conducted on asymptomatic cases for students closely related to the meals service.

**Table 2 ijerph-17-03149-t002:** Univariate analysis of water-contaminated foodstuff exposure associated with the diarrhoeagenic enteropathogenic *Escherichia coli* (*EPEC*) infection outbreak.

Date	Food Items	Exposed			Unexposed			RR (95% CI)	*P-*Value
June 25		Total	Cases	AR (%)	Total	Cases	AR (%)		
	School cafeteria water	87	10	11.5	19	3	15.8	0.73 (0.22–2.4)	0.896
	Purified water	93	11	11.8	13	2	15.4	0.77 (0.19–3.08)	0.932
	Black rice	97	11	11.3	9	2	22.2	0.51 (0.13–1.95)	0.674
	Soybean Paste Stew	87	10	11.5	19	3	15.8	0.73 (0.22–2.34)	0.896
	Apricot ade	72	10	13.9	34	3	8.8	1.57 (0.46–5.35)	0.671
	Pork Belly, pepper paste and Bulgogi	82	9	11.0	24	4	16.7	0.66 (0.22–1.95)	0.694
	Lettuce Wrap with Ssamjang	96	11	11.5	10	2	20.0	0.57 (0.14–2.22)	0.782
	Radish with soybean paste	93	9	9.7	13	4	30.8	0.31 (0.11–0.87)	0.085
	Milk	89	11	12.4	17	2	11.8	1.05 (0.26–4.32)	0.738
June 26	School cafeteria water	91	11	12.1	15	2	13.3	0.90 (0.22–3.69)	0.773
	Purified water	86	11	12.8	20	2	10.0	1.27 (0.30–5.32)	0.972
	Pork barbecue	80	11	13.8	26	2	7.7	1.78 (0.42–7.54)	0.636
	Grilled Tofu with Egg	76	10	13.2	30	3	10.0	1.31 (0.38–4.45)	0.906
	Seasoned green onion and lettuce	71	11	15.5	35	2	5.7	2.71(0.63–11.57)	0.259
	Beef Seaweed Soup	96	11	11.5	10	2	20.0	0.57 (0.14–2.22)	0.782
	Ice mango	96	10	10.4	10	3	30.0	0.34 (0.11–1.05)	0.197
	Nutrition Rice	95	11	11.6	11	2	18.2	0.63 (0.16–2.50)	0.884
	Milk	73	9	12.3	33	4	12.1	1.01 (0.33–3.06)	0.772
June 27	School cafeteria water	93	11	11.8	13	2	15.4	0.76 (0.19–3.08)	0.932
	Purified water	95	11	11.6	11	2	18.2	0.63 (0.16–2.50)	0.884
	Riceballs with Dried Seaweed	89	11	12.4	17	2	11.8	1.05 (0.25–4.32)	0.738
	Fried dumplings with vegetables in spicy sauce	66	10	15.2	40	3	7.5	2.02 (0.59–6.90)	0.391
	boiled egg	80	9	11.3	26	4	15.4	0.73 (0.24–2.17)	0.830
	Watermelon	91	10	11.0	15	3	20.0	0.54 (0.170–1.76)	0.575
	Cold Buckwheat Noodles with radish	90	9	10.0	16	4	25.0	0.40 (0.13–1.14)	0.203
	Cucumber chili with ssamjang	58	11	19.0	48	2	4.2	4.55 (1.05–19.54)	0.044
	Milk	90	9	10.0	16	4	25.0	0.40 (0.13–1.14)	0.203
June 28	School cafeteria water	97	12	12.4	9	1	11.1	1.11 (0.16–7.61)	0.674
	Purified water	100	12	12.0	6	1	16.7	0.72 (0.11–4.65)	0.763
	Barley Rice	88	10	11.4	18	3	16.7	0.68 (0.20–2.23)	0.818
	Soybean paste soup with dried radish leaves	88	10	11.4	18	3	16.7	0.68 (0.20–2.23)	0.818
	Drinking ice	96	10	10.4	10	3	30.0	0.34 (0.11–1.05)	0.197
	Beef quail roe stew	91	11	12.1	15	2	13.3	0.90 (0.22–3.69)	0.773
	Seasoned cucumber and chives	60	12	20.0	46	1	2.2	9.20 (1.24–68.22)	0.013
	Milk	98	10	10.2	8	3	37.5	0.27 (0.09–0.79)	0.089
	Braised eel	97	11	11.3	9	2	22.2	0.51 (0.13–1.95)	0.674
June 29	School cafeteria water	85	10	11.8	21	3	14.3	0.82 (0.24–2.73)	0.955
	Purified water	95	11	11.6	11	2	18.2	0.63 (0.16–2.50)	0.884
	Short Rib Soup	95	10	10.5	11	3	27.3	0.38 (0.12–1.19)	0.264
	Turmeric Rice	96	12	12.5	10	1	10.0	1.25 (0.18–8.64)	0.782
	Steamed sesame leaves	96	12	12.5	10	1	10.0	1.25 (0.18–8.64)	0.782
	Melon	100	11	11.0	6	2	33.3	0.33 (0.09–1.16)	0.328
	Mushroom pancake	97	11	11.3	9	2	22.2	0.51 (0.13–1.95)	0.674
	Milk	97	11	11.3	9	2	22.2	0.51 (0.133–1.95)	0.674
	Stir-fried Pork with tomato	89	11	12.4	17	2	11.8	1.05 (0.25–4.32)	0.738

**Table 3 ijerph-17-03149-t003:** Laboratory tests of the food and environmental specimens.

Specimens	Sample Name	No. of Samples	Test Result (%)	Bacterial Strain
Food Poisoning Bacteria (16 species) and Virus (6 species) *	Preserved food	39	Not detected	
	Knife, cutting board, dishcloth	3	Not detected	
	Drinking water	4	Not detected	
	Cooking water (from village water supply)	1	1(100.0)	*EPEC* O87
	Dishwashing water (from village water supply)	1	1 (100.0)	*EPEC* O87
Drinking waterborne contaminants (9 items) **	Cooking water (from Village water supply)	1	Total coliforms (+), faecal coliforms (+)	
	Drinking water (from Village water supply)	1	Total coliforms (+), faecal coliforms (+)	
	Main school water Purifier (from Village water supply)	1	Total coliforms (+)	
	Boiled water in cafeteria	1	Not detected	
	Purified water in ‘A’ branch school	1	Not detected	
	Purified water in ‘B’ branch school	1	Not detected	

* Bacteria: Cholera, *Salmonella typhi, Salmonella paratyphi*, Shigellosis, enterohemorrhagic *E. coli, Salmonella* spp., *Vibrio parahaemolyticus*, enterotoxic *E. coli*, enteroinvasive *E. coli*, enteropathogenic *E. coli, Campylobacter jejuni, Clostridium perfringens, Staphylococcus aureus, Bacillus cereus, Yersinia enterocolitica*, and *Listeria monocytogenes*; Virus: Hepatitis A, *Rotavirus, Astrovirus, Adenovirus, Norovirus*, and *Sapovirus*. ** Total colonies, total coliforms, faecal coliforms, NH_3_-N, NO_3_-N, residual chlorine, KMnO (consumed), Cl^−^, and SO_4_^2−.^

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
