# Peer review of "A Diarrhoeagenic Enteropathogenic Escherichia coli (EPEC) Infection Outbreak That Occurred among Elementary School Children in Gyeongsangbuk-Do Province of South Korea Was Associated with Consumption of Water-Contaminated Food Items"

_ijerph, 2020, doi:10.3390/ijerph17093149_

Round 1
Reviewer 1 Report
The authors investigated Diarrhoeagenic Enteropathogenic Escherichia coli (EPEC) infection outbreak occurred among elementary school in South Korea. This is an important report for the development of public health.
However, this manuscript is just a report of food poisoning that occurred in a certain region. It is difficult to find that new insights were obtained in this paper. And, such a study need to be reviewed by the Ethics Committee. If the authors do not have any ethical proof, this manuscript is not accepted.
Major comments
- Authors should describe the target elementary school and its surrounding areas. Many readers may think that Korea has many big cities like Seoul and Busan.
- I think authors should further emphasize the significance of this epidemiological study: differences and advantage from other surveys.
Minor comments
Authors write this expression “Diarrhoeagenic Enteropathogenic Escherichia Coli (EPEC)” frequently. Once you have explained the abbreviation, please use it afterwards.
Author Response
First Reviewer-First Round
Journal: International Journal of Environmental Research and Public Health (IJERPH)
Manuscript ID: ijerph-784927
Title: Diarrhoeagenic Enteropathogenic Escherichia coli (EPEC) infection
outbreak occurred among elementary school children in Gyeongbuk-Do Province
of South Korea associated with consumption of water contaminated food items
Comments and Suggestions for Authors
The authors investigated Diarrhoeagenic Enteropathogenic Escherichia coli (EPEC) infection outbreak occurred among elementary school in South Korea. This is an important report for the development of public health.
However, this manuscript is just a report of food poisoning that occurred in a certain region. It is difficult to find that new insights were obtained in this paper. And, such a study need to be reviewed by the Ethics Committee. If the authors do not have any ethical proof, this manuscript is not accepted.
Response: We highly appreciate reviewer’s valuable comments and suggestions. We partially agree that there are not a number of innovative insights in the manuscript, however, it is worthwhile to warrant the stakeholders about the neglected issues of EHEC infections occurred as a result of consumption of contaminated food and water in developed countries especially in non-community settings. In regard to the ethics, we have clearly mentioned it adding a separate topic in the methods section of the revised version of the manuscript. In addition, we will submit the same evidence to the journal editorial office along with revision. Also, we have revised the manuscript extensively based on reviewers’ comments and suggestions. All changes are marked with blue colored writing wherever changes have been made in order to allow reviewers’ verifications.
Major comments
Comment: 1. Authors should describe the target elementary school and its surrounding areas. Many readers may think that Korea has many big cities like Seoul and Busan.
Response: Agree. We have discussed about the study setting in the current version.
Comment: 2. I think authors should further emphasize the significance of this epidemiological study: differences and advantages from other surveys.
Response: Agree. The significance of the study has been written in the concluding paragraph of the introduction.
Minor comments
Comment: Authors write this expression “Diarrhoeagenic Enteropathogenic Escherichia Coli (EPEC)” frequently. Once you have explained the abbreviation, please use it afterward.
Response: Agree. We greatly appreciate the review’s comment. It has been well addressed now.
Reviewer 2 Report
This study describes an outbreak of gastrointestinal illness in schoolchildren after ingestion of water contaminated food items. The handling is of routine character which any infectious disease control unit will do according to guidelines using routine laboratory diagnostic. It is executed according to guidelines with the exception that data are not gender divided. To find out if women, men, boys and girls are differently affected would have added a dimension of interest, and of possibility of prevention to the paper.It says in the Introduction tha tEHEC is a common food borne pathogen. Considering the high mortality and morbidity this is a doubtful statement and the authors should look more into this.
Author Response
Second Reviewer-First Round
Journal: International Journal of Environmental Research and Public Health (IJERPH)
Manuscript ID: ijerph-784927
Title: Diarrhoeagenic Enteropathogenic Escherichia coli (EPEC) infection
outbreak occurred among elementary school children in Gyeongbuk-Do Province
of South Korea associated with consumption of water contaminated food items
Comments and Suggestions for Authors
This study describes an outbreak of gastrointestinal illness in schoolchildren after the ingestion of water contaminated food items. The handling is of routine character which any infectious disease control unit will do according to guidelines using routine laboratory diagnostic.
Response: We thank you very much and highly appreciate the reviewer’s valuable comments and suggestions. We have revised the manuscript based on reviewers’ comments and suggestions. All changes are marked with blue colored writing in this revised version of the manuscript to allow reviewers’ verifications.
Comment: It is executed according to guidelines with the exception that data are not gendered divided. To find out if women, men, boys, and girls are differently affected would have added a dimension of interest, and of the possibility of prevention to the paper.
Response: Agree. It has been addressed as suggested in the descriptive result part of the manuscript.
Comment: It says in the Introduction that EHEC is a common foodborne pathogen. Considering the high mortality and morbidity this is a doubtful statement and the authors should look more into this.
Response: Absolutely agree. We have now modified the sentence in the concluding paragraph of the introduction so that this does not contradict the mortality and morbidity statistics mentioned in the early part of the introduction.
Reviewer 3 Report
Dear Editorial Board IJERPH
Regarding the manuscript entitled "Diarrhoeagenic Enteropathogenic Escherichia coli (EPEC) infection outbreak occurred among elementary school children in Gyeongbuk-Do Province of South Korea associated with consumption of water contaminated food item" I read it and my observations are stated bellow.
- It is a good manuscript, with minor corrections.
- However, I have not seen new findings, which can be cited in a near future. It seems a case study.
- In some phrases I noted a certain slouch with standards of writing as in line 9 and line 294.
- So, I recommend a strong revision regarding style and standards.
Minors:
- L2 and throughout the manuscript: genus and species must be write in italicized way.
- L83 "in a nutshell" is quite informal.
- L142 p<=0.05 instead of p<0.05.
- L138 "nitrate-nitrogen" and "ammonia-nitrogen".
Yours sincerely.
Author Response
Third Reviewer-First Round
Journal: International Journal of Environmental Research and Public Health (IJERPH)
Manuscript ID: ijerph-784927
Title: Diarrhoeagenic Enteropathogenic Escherichia coli (EPEC) infection
outbreak occurred among elementary school children in Gyeongbuk-Do Province
of South Korea associated with consumption of water contaminated food items
Comments and Suggestions for Authors
Dear Editorial Board IJERPH
Regarding the manuscript entitled "Diarrhoeagenic Enteropathogenic Escherichia coli (EPEC) infection outbreak occurred among elementary school children in Gyeongbuk-Do Province of South Korea associated with consumption of water contaminated food item" I read it and my observations are stated bellow.
Response: Thank you very much for the encouragement. We have revised the manuscript extensively based on reviewers’ comments and suggestions. All changes are marked with blue colored writing in this revised version of the manuscript to allow reviewers’ verifications.
Major
Comments: It is a good manuscript, with minor corrections. However, I have not seen new findings, which can be cited in a near future. It seems a case study.
Response: Thank you very much for the appreciation and suggestions. We partially agree that there are not a number of innovative insights in the manuscript, however, it is worthwhile to warrant the stakeholders about the neglected issues of EHEC infections occurred as a result of the consumption of contaminated food and water in developed countries especially in non-community settings.
Comments: In some phrases, I noted a certain slouch with standards of writing as in line 9 and line 294. So, I recommend a strong revision regarding style and standards.
Response: Agree. We guess it might be the typos made during PDF version conversion by the Editorial office before it is being submitted to the reviewers. However, this has been corrected now.
Minors:
Comment: L2 and throughout the manuscript: genus and species must be written in an italicized way.
Response: Agree. Checked and corrected throughout the manuscript.
Comment: L83 "in a nutshell" is quite informal.
Response: Agree. Corrected
Comment: L142 p<=0.05 instead of p<0.05.
Response: Agree. Corrected
Comment: L138 "nitrate-nitrogen" and "ammonia-nitrogen".
Response: Agree. Corrected.
Reviewer 4 Report
I appreciate a lot the manuscript, well written and well thought.
For me it is a manuscript to be published and it is already ready to be published
Only some suggestions:
- Introduction: well done. I prefer if you can use a few words on Escherichia coil (gram-negative bacteria, serotypes, ...) and the global burden of diseases
- Methods: well written
- Results: very interesting and is important to share this experience
- Discussion and Conclusion: very very well done. If you are agree to add at references this article already publishes on this journal: "Marotta C, et al. The At-Risk Child Clinic (ARCC): 3 Years of Health Activities in Support of the Most Vulnerable Children in Beira, Mozambique. Int J Environ Res Public Health. 2018;15(7):1350" to underline how children are a most vulnerable part and also scientific research represents a strategic point to better understand health demands and to monitor the quality of care provided to this vulnerable population group
Author Response
Fourth Reviewer-First Round
Journal: International Journal of Environmental Research and Public Health (IJERPH)
Manuscript ID: ijerph-784927
Title: Diarrhoeagenic Enteropathogenic Escherichia coli (EPEC) infection
outbreak occurred among elementary school children in Gyeongbuk-Do Province
of South Korea associated with consumption of water contaminated food items
Comments and Suggestions for Authors
I appreciate a lot the manuscript, well written and well thought. For me it is a manuscript to be published and it is already ready to be published
Response: Thank you very much for your encouragement. We have made some revisions in the manuscript based on reviewers’ comments and suggestions. All changes are marked with blue colored writing in the revised version of the manuscript to allow reviewers’ verifications.
Only some suggestions:
Comment: 1. Introduction: well done. I prefer if you can use a few words on Escherichia coil (gram-negative bacteria, serotypes, ...) and the global burden of diseases
Response: Agree. Corrected
Comment: 2. Methods: well written
Response: Thank you very much for your appreciation.
Comment: 3. Results: very interesting and is important to share this experience
Response: Thank you very much for your appreciation.
Comment: 4. Discussion and Conclusion: very very well done. If you are agree to add at references this article already publishes on this journal: "Marotta C, et al. The At-Risk Child Clinic (ARCC): 3 Years of Health Activities in Support of the Most Vulnerable Children in Beira, Mozambique. Int J Environ Res Public Health. 2018;15(7):1350" to underline how children are a most vulnerable part and also scientific research represents a strategic point to better understand health demands and to monitor the quality of care provided to this vulnerable population group
Response: Thank you very much for your appreciation and great suggestions. We have now cited the same paper in the discussion section of the manuscript.
Round 2
Reviewer 1 Report
The author has properly corrected the issue I pointed out. I think this manuscript has been improved and becomes very meaningful. It seems to be worth being published in International Journal of Environmental Research and Public Health.